# SGD Learns the Conjugate Kernel Class of the Network

**Amit Daniely**
Hebrew University and Google Research
`amit.daniely@mail.huji.ac.il`

### Abstract

We show that the standard stochastic gradient decent (SGD) algorithm is guaranteed to learn, in polynomial time, a function that is competitive with the best function in the conjugate kernel space of the network, as defined in Daniely et al. [2016]. The result holds for log-depth networks from a rich family of architectures. To the best of our knowledge, it is the first polynomial-time guarantee for the standard neural network learning algorithm for networks of depth more that two.

As corollaries, it follows that for neural networks of any depth between 2 and $\log(n)$, SGD is guaranteed to learn, in polynomial time, constant degree polynomials with polynomially bounded coefficients. Likewise, it follows that SGD on large enough networks can learn any continuous function (not in polynomial time), complementing classical expressivity results.

## 1 Introduction

While stochastic gradient decent (SGD) from a random initialization is probably the most popular supervised learning algorithm today, we have very few results that depicts conditions that guarantee its success. Indeed, to the best of our knowledge, Andoni et al. [2014] provides the only known result of this form, and it is valid in a rather restricted setting. Namely, for depth-2 networks, where the underlying distribution is Gaussian, the algorithm is full gradient decent (rather than SGD), and the task is regression when the learnt function is a constant degree polynomial.

We build on the framework of Daniely et al. [2016] to establish guarantees on SGD in a rather general setting. Daniely et al. [2016] defined a framework that associates a reproducing kernel to a network architecture. They also connected the kernel to the network via the random initialization. Namely, they showed that right after the random initialization, any function in the kernel space can be approximated by changing the weights of the last layer. The quality of the approximation depends on the size of the network and the norm of the function in the kernel space.

As optimizing the last layer is a convex procedure, the result of Daniely et al. [2016] intuitively shows that the optimization process starts from a favourable point for learning a function in the conjugate kernel space. In this paper we verify this intuition. Namely, for a fairly general family of architectures (that contains fully connected networks and convolutional networks) and supervised learning tasks, we show that if the network is large enough, the learning rate is small enough, and the number of SGD steps is large enough as well, SGD is guaranteed to learn any function in the corresponding kernel space. We emphasize that the number of steps and the size of the network are only required to be polynomial (which is best possible) in the relevant parameters – the norm of the function, the required accuracy parameter ($\epsilon$), and the dimension of the input and the output of the network. Likewise, the result holds for any input distribution.

To evaluate our result, one should understand which functions it guarantee that SGD will learn. Namely, what functions reside in the conjugate kernel space, how rich it is, and how good those functions are as predictors. From an empirical perspective, in [Daniely et al., 2017], it is shown that for standard convolutional networks the conjugate class contains functions whose performance is

close to the performance of the function that is actually learned by the network. This is based on experiments on the standard CIFAR-10 dataset. From a theoretical perspective, we list below a few implications that demonstrate the richness of the conjugate kernel space. These implications are valid for fully connected networks of any depth between 2 and $\log(n)$, where $n$ is the input dimension. Likewise, they are also valid for convolutional networks of any depth between 2 and $\log(n)$, and with constantly many convolutional layers.

- SGD is guaranteed to learn in polynomial time constant degree polynomials with polynomially bounded coefficients. As a corollary, SGD is guaranteed to learn in polynomial time conjunctions, DNF and CNF formulas with constantly many terms, and DNF and CNF formulas with constantly many literals in each term. These function classes comprise a considerable fraction of the function classes that are known to be poly-time (PAC) learnable by *any method*. Exceptions include constant degree polynomial thresholds with no restriction on the coefficients, decision lists and parities.

- SGD is guaranteed to learn, not necessarily in polynomial time, *any* continuous function. This complements classical universal approximation results that show that neural networks can *(approximately) express* any continuous function (see Scarselli and Tsoi [1998] for a survey). Our results strengthen those results and show that networks are not only able to express those functions, but actually guaranteed to *learn* them.

## 1.1 Related work

**Guarantees on SGD.** As noted above, there are very few results that provide polynomial time guarantees for SGD on NN. One notable exception is the work of Andoni et al. [2014], that proves a result that is similar to ours, but in a substantially more restricted setting. Concretely, their result holds for depth-2 fully connected networks, as opposed to rather general architecture and constant or logarithmic depth in our case. Likewise, the marginal distribution on the instance space is assumed to be Gaussian or uniform, as opposed to arbitrary in our case. In addition, the algorithm they consider is full gradient decent, which corresponds to SGD with infinitely large mini-batch, as opposed to SGD with arbitrary mini-batch size in our case. Finally, the underlying task is regression in which the target function is a constant degree polynomial, whereas we consider rather general supervised learning setting.

**Other polynomial time guarantees on learning deep architectures.** Various recent papers show that poly-time learning is possible in the case that the the learnt function can be realized by a neural network with certain (usually fairly strong) restrictions on the weights [Livni et al., 2014, Zhang et al., 2016a, 2015, 2016b], or under the assumption that the data is generated by a generative model that is derived from the network architecture [Arora et al., 2014, 2016]. We emphasize that the main difference of those results from our results and the results of Andoni et al. [2014] is that they do not provide guarantees on the standard SGD learning algorithm. Rather, they show that under those aforementioned conditions, there are *some algorithms*, usually very different from SGD on the network, that are able to learn in polynomial time.

**Connection to kernels.** As mentioned earlier, our paper builds on Daniely et al. [2016], who developed the association of kernels to NN which we rely on. Several previous papers [Mairal et al., 2014, Cho and Saul, 2009, Rahimi and Recht, 2009, 2007, Neal, 2012, Williams, 1997, Kar and Karnick, 2012, Pennington et al., 2015, Bach, 2015, 2014, Hazan and Jaakkola, 2015, Anselmi et al., 2015] investigated such associations, but in a more restricted settings (i.e., for less architectures). Some of those papers [Rahimi and Recht, 2009, 2007, Daniely et al., 2016, Kar and Karnick, 2012, Bach, 2015, 2014] also provide measure of concentration results, that show that w.h.p. the random initialization of the network's weights is reach enough to approximate the functions in the corresponding kernel space. As a result, these papers provide polynomial time guarantees on the variant of SGD, where only the last layer is trained. We remark that with the exception of Daniely et al. [2016], those results apply just to depth-2 networks.

## 1.2 Discussion and future directions

We next want to place this work in the appropriate learning theoretic context, and to elaborate further on this paper's approach for investigating neural networks. For the sake of concreteness, let us

restrict the discussion to binary classification over the Boolean cube. Namely, given examples from a distribution $\mathcal{D}$ on $\{\pm 1\}^n \times \{0, 1\}$, the goal is to learn a function $h : \{\pm 1\}^n \to \{0, 1\}$ whose 0-1 error, $\mathcal{L}_{\mathcal{D}}^{0-1}(h) = \Pr_{(\mathbf{x},y) \sim \mathcal{D}}(h(\mathbf{x}) \neq y)$, is as small as possible. We will use a bit of terminology. A *model* is a distribution $\mathcal{D}$ on $\{\pm 1\}^n \times \{0, 1\}$ and a *model class* is a collection $\mathcal{M}$ of models. We note that any function class $\mathcal{H} \subset \{0, 1\}^{\{\pm 1\}^n}$ defines a model class, $\mathcal{M}(\mathcal{H})$, consisting of all models $\mathcal{D}$ such that $\mathcal{L}_{\mathcal{D}}^{0-1}(h) = 0$ for some $h \in \mathcal{H}$. We define the *capacity* of a model class as the minimal number $m$ for which there is an algorithm such that for every $\mathcal{D} \in \mathcal{M}$ the following holds. Given $m$ samples from $\mathcal{D}$, the algorithm is guaranteed to return, w.p. $\geq \frac{9}{10}$ over the samples and its internal randomness, a function $h : \{\pm 1\}^n \to \{0, 1\}$ with 0-1 error $\leq \frac{1}{10}$. We note that for function classes the capacity is the VC dimension, up to a constant factor.

Learning theory analyses learning algorithms via model classes. Concretely, one fixes some model class $\mathcal{M}$ and show that the algorithm is guaranteed to succeed whenever the underlying model is from $\mathcal{M}$. Often, the connection between the algorithm and the class at hand is very clear. For example, in the case that the model is derived from a function class $\mathcal{H}$, the algorithm might simply be one that finds a function in $\mathcal{H}$ that makes no mistake on the given sample. The natural choice for a model class for analyzing SGD on NN would be the class of all functions that can be realized by the network, possibly with some reasonable restrictions on the weights. Unfortunately, this approach it is probably doomed to fail, as implied by various computational hardness results [Blum and Rivest, 1989, Kearns and Valiant, 1994, Blum et al., 1994, Kharitonov, 1993, Klivans and Sherstov, 2006, 2007, Daniely et al., 2014, Daniely and Shalev-Shwartz, 2016].

So, what model classes should we consider? With a few isolated exceptions (e.g. Bshouty et al. [1998]) all known efficiently learnable model classes are either a linear model class, or contained in an efficiently learnable linear model class. Namely, functions classes composed of compositions of some predefined embedding with linear threshold functions, or linear functions over some finite field.

Coming up we new tractable models would be a fascinating progress. Still, as linear function classes are the main tool that learning theory currently has for providing guarantees on learning, it seems natural to try to analyze SGD via linear model classes. Our work follows this line of thought, and we believe that there is much more to achieve via this approach. Concretely, while our bounds are polynomial, the degree of the polynomials is rather large, and possibly much better quantitative bounds can be achieved. To be more concrete, suppose that we consider simple fully connected architecture, with 2-layers, ReLU activation, and $n$ hidden neurons. In this case, the capacity of the model class that our results guarantee that SGD will learn is $\Theta\left(n^{\frac{1}{3}}\right)$. For comparison, the capacity of the class of all functions that are realized by this network is $\Theta\left(n^2\right)$. As a challenge, we encourage the reader to prove that with this architecture (possibly with an activation that is different from the ReLU), SGD is guaranteed to learn *some* model class of capacity that is super-linear in $n$.

## 2  Preliminaries

**Notation.**  We denote vectors by bold-face letters (e.g. $\mathbf{x}$), matrices by upper case letters (e.g. $W$), and collection of matrices by bold-face upper case letters (e.g. $\mathbf{W}$). The $p$-norm of $\mathbf{x} \in \mathbb{R}^d$ is denoted by $\|\mathbf{x}\|_p = \left(\sum_{i=1}^d |x_i|^p\right)^{\frac{1}{p}}$. We will also use the convention that $\|\mathbf{x}\| = \|\mathbf{x}\|_2$. For functions $\sigma : \mathbb{R} \to \mathbb{R}$ we let

$$\|\sigma\| := \sqrt{\mathbb{E}_{X \sim \mathcal{N}(0,1)} \sigma^2(X)} = \sqrt{\frac{1}{\sqrt{2\pi}} \int_{-\infty}^{\infty} \sigma^2(x) e^{-\frac{x^2}{2}} \, dx}.$$

Let $G = (V, E)$ be a directed acyclic graph. The set of neighbors incoming to a vertex $v$ is denoted $\text{in}(v) := \{u \in V \mid uv \in E\}$. We also denote $\deg(v) = |\text{in}(v)|$. Given weight function $\delta : V \to [0, \infty)$ and $U \subset V$ we let $\delta(U) = \sum_{u \in U} \delta(u)$. The $d - 1$ dimensional sphere is denoted $\mathbb{S}^{d-1} = \{\mathbf{x} \in \mathbb{R}^d \mid \|\mathbf{x}\| = 1\}$. We use $[x]_+$ to denote $\max(x, 0)$.

**Input space.**  Throughout the paper we assume that each example is a sequence of $n$ elements, each of which is represented as a unit vector. Namely, we fix $n$ and take the input space to be $\mathcal{X} = \mathcal{X}_{n,d} = \left(\mathbb{S}^{d-1}\right)^n$. Each input example is denoted,

$$\mathbf{x} = (\mathbf{x}^1, \ldots, \mathbf{x}^n), \text{ where } \mathbf{x}^i \in \mathbb{S}^{d-1}. \tag{1}$$

While this notation is slightly non-standard, it unifies input types seen in various domains (see Daniely et al. [2016]).

**Supervised learning.** The goal in supervised learning is to devise a mapping from the input space $\mathcal{X}$ to an output space $\mathcal{Y}$ based on a sample $S = \{(\mathbf{x}_1, y_1), \ldots, (\mathbf{x}_m, y_m)\}$, where $(\mathbf{x}_i, y_i) \in \mathcal{X} \times \mathcal{Y}$ drawn i.i.d. from a distribution $\mathcal{D}$ over $\mathcal{X} \times \mathcal{Y}$. A supervised learning problem is further specified by an output length $k$ and a loss function $\ell : \mathbb{R}^k \times \mathcal{Y} \to [0, \infty)$, and the goal is to find a predictor $h : \mathcal{X} \to \mathbb{R}^k$ whose loss, $\mathcal{L}_\mathcal{D}(h) := \mathbb{E}_{(\mathbf{x},y)\sim\mathcal{D}} \, \ell(h(\mathbf{x}), y)$, is small. The *empirical* loss $\mathcal{L}_S(h) := \frac{1}{m} \sum_{i=1}^m \ell(h(\mathbf{x}_i), y_i)$ is commonly used as a proxy for the loss $\mathcal{L}_\mathcal{D}$. When $h$ is defined by a vector $\mathbf{w}$ of parameters, we will use the notations $\mathcal{L}_\mathcal{D}(\mathbf{w}) = \mathcal{L}_\mathcal{D}(h)$, $\mathcal{L}_S(\mathbf{w}) = \mathcal{L}_S(h)$ and $\ell_{(\mathbf{x},y)}(\mathbf{w}) = \ell(h(\mathbf{x}), y)$.

Regression problems correspond to $k = 1$, $\mathcal{Y} = \mathbb{R}$ and, for instance, the squared loss $\ell^{\text{square}}(\hat{y}, y) = (\hat{y} - y)^2$. Binary classification is captured by $k = 1$, $\mathcal{Y} = \{\pm 1\}$ and, say, the zero-one loss $\ell^{0-1}(\hat{y}, y) = \mathbf{1}[\hat{y}y \leq 0]$ or the hinge loss $\ell^{\text{hinge}}(\hat{y}, y) = [1 - \hat{y}y]_+$. Multiclass classification is captured by $k$ being the number of classes, $\mathcal{Y} = [k]$, and, say, the zero-one loss $\ell^{0-1}(\hat{y}, y) = \mathbf{1}[\hat{y}_y \leq \text{argmax}_{y'} \hat{y}_{y'}]$ or the logistic loss $\ell^{\log}(\hat{\mathbf{y}}, y) = -\log(p_y(\hat{\mathbf{y}}))$ where $\mathbf{p} : \mathbb{R}^k \to \Delta^{k-1}$ is given by $p_i(\hat{\mathbf{y}}) = \frac{e^{\hat{y}_i}}{\sum_{j=1}^k e^{\hat{y}_j}}$. A loss $\ell$ is $L$-Lipschitz if for all $y \in \mathcal{Y}$, the function $\ell_y(\hat{y}) := \ell(\hat{y}, y)$ is $L$-Lipschitz. Likewise, it is convex if $\ell_y$ is convex for every $y \in \mathcal{Y}$.

**Neural network learning.** We define a *neural network* $\mathcal{N}$ to be a vertices weighted directed acyclic graph (DAG) whose nodes are denoted $V(\mathcal{N})$ and edges $E(\mathcal{N})$. The weight function will be denoted by $\delta : V(\mathcal{N}) \to [0, \infty)$, and its sole role would be to dictate the distribution of the initial weights. We will refer $\mathcal{N}$'s nodes by *neurons*. Each of non-input neuron, i.e. neuron with incoming edges, is associated with an *activation* function $\sigma_v : \mathbb{R} \to \mathbb{R}$. In this paper, an activation can be any function $\sigma : \mathbb{R} \to \mathbb{R}$ that is right and left differentiable, square integrable with respect to the Gaussian measure on $\mathbb{R}$, and is *normalized* in the sense that $\|\sigma\| = 1$. The set of neurons having only incoming edges are called the output neurons. To match the setup of supervised learning defined above, a network $\mathcal{N}$ has $nd$ input neurons and $k$ output neurons, denoted $o_1, \ldots, o_k$. A network $\mathcal{N}$ together with a weight vector $\mathbf{w} = \{w_{uv} \mid uv \in E\} \cup \{b_v \mid v \in V \text{ is an internal neuron}\}$ defines a predictor $h_{\mathcal{N},\mathbf{w}} : \mathcal{X} \to \mathbb{R}^k$ whose prediction is given by "propagating" $\mathbf{x}$ forward through the network. Concretely, we define $h_{v,\mathbf{w}}(\cdot)$ to be the output of the subgraph of the neuron $v$ as follows: for an input neuron $v$, $h_{v,\mathbf{w}}$ outputs the corresponding coordinate in $\mathbf{x}$, and internal neurons, we define $h_{v,\mathbf{w}}$ recursively as

$$h_{v,\mathbf{w}}(\mathbf{x}) = \sigma_v \left( \sum_{u \in \text{in}(v)} w_{uv} h_{u,\mathbf{w}}(\mathbf{x}) + b_v \right).$$

For output neurons, we define $h_{v,\mathbf{w}}$ as

$$h_{v,\mathbf{w}}(\mathbf{x}) = \sum_{u \in \text{in}(v)} w_{uv} h_{u,\mathbf{w}}(\mathbf{x}).$$

Finally, we let $h_{\mathcal{N},\mathbf{w}}(\mathbf{x}) = (h_{o_1,\mathbf{w}}(\mathbf{x}), \ldots, h_{o_k,\mathbf{w}}(\mathbf{x}))$.

We next describe the learning algorithm that we analyze in this paper. While there is no standard training algorithm for neural networks, the algorithms used in practice are usually quite similar to the one we describe, both in the way the weights are initialized and the way they are updated. We will use the popular Xavier initialization [Glorot and Bengio, 2010] for the network weights. Fix $0 \leq \beta \leq 1$. We say that $\mathbf{w}^0 = \{w_{uv}^0\}_{uv \in E} \cup \{b_v\}_{v \in V \text{ is an internal neuron}}$ are $\beta$-*biased random weights* (or, $\beta$-biased random initialization) if each weight $w_{uv}$ is sampled independently from a normal distribution with mean 0 and variance $(1 - \beta)d\delta(u)/\delta(\text{in}(v))$ if $u$ is an input neuron and $(1 - \beta)\delta(u)/\delta(\text{in}(v))$ otherwise. Finally, each bias term $b_v$ is sampled independently from a normal distribution with mean 0 and variance $\beta$. We note that the rational behind this initialization scheme is that for every example $\mathbf{x}$ and every neuron $v$ we have $\mathbb{E}_{\mathbf{w}_0} \left( h_{v,\mathbf{w}_0}(\mathbf{x}) \right)^2 = 1$ (see Glorot and Bengio [2010])

**Kernel classes.** A function $\kappa : \mathcal{X} \times \mathcal{X} \to \mathbb{R}$ is a *reproducing kernel*, or simply a kernel, if for every $\mathbf{x}_1, \ldots, \mathbf{x}_r \in \mathcal{X}$, the $r \times r$ matrix $\Gamma_{i,j} = \{\kappa(\mathbf{x}_i, \mathbf{x}_j)\}$ is positive semi-definite. Each kernel induces a Hilbert space $\mathcal{H}_\kappa$ of functions from $\mathcal{X}$ to $\mathbb{R}$ with a corresponding norm $\|\cdot\|_\kappa$. For $\mathbf{h} \in \mathcal{H}_\kappa^k$ we denote $\|\mathbf{h}\|_\kappa = \sqrt{\sum_{i=1}^k \|h_i\|_\kappa^2}$. A kernel and its corresponding space are *normalized* if $\forall \mathbf{x} \in \mathcal{X}, \, \kappa(\mathbf{x}, \mathbf{x}) = 1$.

---

**Algorithm 1** Generic Neural Network Training

---

**Input:** Network $\mathcal{N}$, learning rate $\eta > 0$, batch size $m$, number of steps $T > 0$, bias parameter $0 \leq \beta \leq 1$, flag zero_prediction_layer $\in \{\text{True}, \text{False}\}$.
Let $\mathbf{w}^0$ be $\beta$-biased random weights
**if** zero_prediction_layer **then**
    Set $w_{uv}^0 = 0$ whenever $v$ is an output neuron
**end if**
**for** $t = 1, \ldots, T$ **do**
    Obtain a mini-batch $S_t = \{(\mathbf{x}_i^t, y_i^t)\}_{i=1}^m \sim \mathcal{D}^m$
    Using back-propagation, calculate a stochastic gradient $\mathbf{v}^t = \nabla \mathcal{L}_{S_t}(\mathbf{w}^t)$
    Update $\mathbf{w}^{t+1} = \mathbf{w}^t - \eta \mathbf{v}^t$
**end for**

---

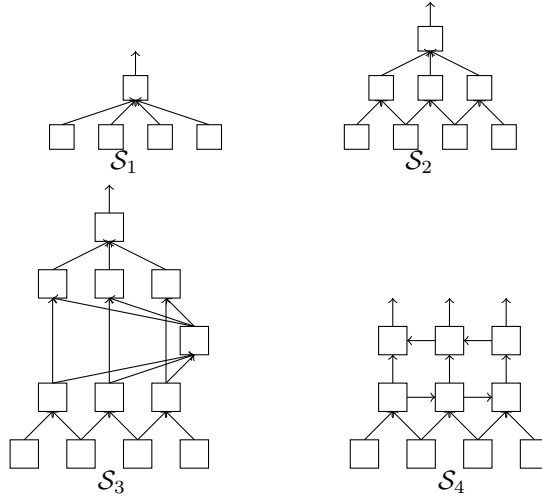

Figure 1: Examples of computation skeletons.

Kernels give rise to popular benchmarks for learning algorithms. Fix a normalized kernel $\kappa$ and $M > 0$. It is well known that that for $L$-Lipschitz loss $\ell$, the SGD algorithm is guaranteed to return a function $\mathbf{h}$ such that $\mathbb{E}\,\mathcal{L}_{\mathcal{D}}(\mathbf{h}) \leq \min_{\mathbf{h}' \in \mathcal{H}_\kappa^k,\, \|\mathbf{h}'\|_\kappa \leq M} \mathcal{L}_{\mathcal{D}}(\mathbf{h}') + \epsilon$ using $\left(\frac{LM}{\epsilon}\right)^2$ examples. In the context of multiclass classification, for $\gamma > 0$ we define $\ell^\gamma : \mathbb{R}^k \times [k] \to \mathbb{R}$ by $\ell^\gamma(\hat{y}, y) = \mathbf{1}[\hat{y}_y \leq \gamma + \max_{y' \neq y} \hat{y}_{y'}]$. We say that a distribution $\mathcal{D}$ on $\mathcal{X} \times [k]$ is $M$-separable w.r.t. $\kappa$ if there is $\mathbf{h}^* \in \mathcal{H}_\kappa^k$ such that $\|\mathbf{h}^*\|_\kappa \leq M$ and $\mathcal{L}_{\mathcal{D}}^1(\mathbf{h}^*) = 0$. In this case, the perceptron algorithm is guaranteed to return a function $\mathbf{h}$ such that $\mathbb{E}\,\mathcal{L}_{\mathcal{D}}^{0-1}(\mathbf{h}) \leq \epsilon$ using $\frac{2M^2}{\epsilon}$ examples. We note that both for perceptron and SGD, the above mentioned results are best possible, in the sense that any algorithm with the same guarantees, will have to use at least the same number of examples, up to a constant factor.

**Computation skeletons [Daniely et al., 2016]**   In this section we define a simple structure which we term a computation skeleton. The purpose of a computational skeleton is to compactly describe a feed-forward computation from an input to an output. A single skeleton encompasses a family of neural networks that share the same skeletal structure. Likewise, it defines a corresponding normalized kernel.

**Definition 1.** *A computation skeleton $\mathcal{S}$ is a DAG with $n$ inputs, whose non-input nodes are labeled by activations, and has a single output node* $\text{out}(\mathcal{S})$.

Figure 1 shows four example skeletons, omitting the designation of the activation functions. We denote by $|\mathcal{S}|$ the number of non-input nodes of $\mathcal{S}$. The following definition shows how a skeleton, accompanied with a replication parameter $r \geq 1$ and a number of output nodes $k$, induces a neural network architecture.

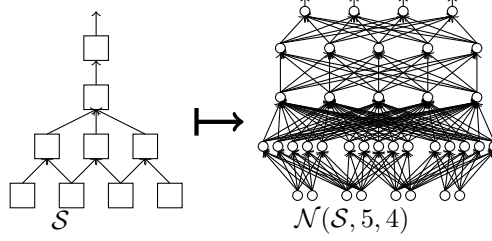

Figure 2: A $(5, 4)$-realization of the computation skeleton $\mathcal{S}$ with $d = 2$.

**Definition 2** (Realization of a skeleton)**.** *Let $\mathcal{S}$ be a computation skeleton and consider input coordinates in $\mathbb{S}^{d-1}$ as in (1). For $r, k \geq 1$ we define the following neural network $\mathcal{N} = \mathcal{N}(\mathcal{S}, r, k)$. For each input node in $\mathcal{S}$, $\mathcal{N}$ has $d$ corresponding input neurons with weight $1/d$. For each internal node $v \in \mathcal{S}$ labelled by an activation $\sigma$, $\mathcal{N}$ has $r$ neurons $v^1, \ldots, v^r$, each with an activation $\sigma$ and weight $1/r$. In addition, $\mathcal{N}$ has $k$ output neurons $o_1, \ldots, o_k$ with the identity activation $\sigma(x) = x$ and weight $1$. There is an edge $v^i u^j \in E(\mathcal{N})$ whenever $uv \in E(\mathcal{S})$. For every output node $v$ in $\mathcal{S}$, each neuron $v^j$ is connected to all output neurons $o_1, \ldots, o_k$. We term $\mathcal{N}$ the $(r, k)$-fold realization of $\mathcal{S}$.*

Note that the notion of the replication parameter $r$ corresponds, in the terminology of convolutional networks, to the number of channels taken in a convolutional layer and to the number of hidden neurons taken in a fully-connected layer.

In addition to networks' architectures, a computation skeleton $\mathcal{S}$ also defines a normalized kernel $\kappa_{\mathcal{S}} : \mathcal{X} \times \mathcal{X} \to [-1, 1]$. To define the kernel, we use the notion of a *conjugate activation*. For $\rho \in [-1, 1]$, we denote by $N_{\rho}$ the multivariate Gaussian distribution on $\mathbb{R}^2$ with mean $0$ and covariance matrix $\left( \begin{smallmatrix} 1 & \rho \\ \rho & 1 \end{smallmatrix} \right)$.

**Definition 3** (Conjugate activation)**.** *The* conjugate activation *of an activation $\sigma$ is the function $\hat{\sigma} : [-1, 1] \to \mathbb{R}$ defined as $\hat{\sigma}(\rho) = \mathbb{E}_{(X,Y) \sim N_{\rho}} \sigma(X)\sigma(Y)$.*

The following definition gives the kernel corresponding to a skeleton

**Definition 4** (Compositional kernels)**.** *Let $\mathcal{S}$ be a computation skeleton and let $0 \leq \beta \leq 1$. For every node $v$, inductively define a kernel $\kappa_v^{\beta} : \mathcal{X} \times \mathcal{X} \to \mathbb{R}$ as follows. For an input node $v$ corresponding to the $i$th coordinate, define $\kappa_v^{\beta}(\mathbf{x}, \mathbf{y}) = \langle \mathbf{x}^i, \mathbf{y}^i \rangle$. For a non-input node $v$, define*

$$\kappa_v^{\beta}(\mathbf{x}, \mathbf{y}) = \hat{\sigma}_v \left( (1 - \beta) \frac{\sum_{u \in \mathrm{in}(v)} \kappa_u^{\beta}(\mathbf{x}, \mathbf{y})}{|\mathrm{in}(v)|} + \beta \right) .$$

*The final kernel $\kappa_{\mathcal{S}}^{\beta}$ is $\kappa_{\mathrm{out}(\mathcal{S})}^{\beta}$. The resulting Hilbert space and norm are denoted $\mathcal{H}_{\mathcal{S}, \beta}$ and $\| \cdot \|_{\mathcal{S}, \beta}$ respectively.*

## 3  Main results

An activation $\sigma : \mathbb{R} \to \mathbb{R}$ is called *$C$-bounded* if $\|\sigma\|_{\infty}, \|\sigma'\|_{\infty}, \|\sigma''\|_{\infty} \leq C$. Fix a skeleton $\mathcal{S}$ and 1-Lipschitz[1] convex loss $\ell$. Define $\mathrm{comp}(\mathcal{S}) = \prod_{i=1}^{\mathrm{depth}(\mathcal{S})} \max_{v \in \mathcal{S}, \mathrm{depth}(v) = i} (\deg(v) + 1)$ and $\mathcal{C}(\mathcal{S}) = (8C)^{\mathrm{depth}(\mathcal{S})} \sqrt{\mathrm{comp}(\mathcal{S})}$, where $C$ is the minimal number for which all the activations in $\mathcal{S}$ are $C$-bounded, and $\mathrm{depth}(v)$ is the maximal length of a path from an input node to $v$. We also define $\mathcal{C}'(\mathcal{S}) = (4C)^{\mathrm{depth}(\mathcal{S})} \sqrt{\mathrm{comp}(\mathcal{S})}$, where $C$ is the minimal number for which all the activations in $\mathcal{S}$ are $C$-Lipschitz and satisfy $|\sigma(0)| \leq C$. Through this and remaining sections we use $\gtrsim$ to hide universal constants. Likewise, we fix the bias parameter $\beta$ and therefore omit it from the relevant notation.

We note that for constant depth skeletons with maximal degree that is polynomial in $n$, $\mathcal{C}(\mathcal{S})$ and $\mathcal{C}'(\mathcal{S})$ are polynomial in $n$. These quantities are polynomial in $n$ also for various log-depth skeletons. For example, this is true for fully connected skeletons, or more generally, layered skeletons with constantly many layers that are not fully connected.

**Theorem 1.** *Suppose that all activations are $C$-bounded. Let $M, \epsilon > 0$. Suppose that we run algorithm 1 on the network $\mathcal{N}(\mathcal{S}, r, k)$ with the following parameters:*

- $\eta = \frac{\eta'}{r}$ *for* $\eta' \lesssim \frac{\epsilon}{(\mathcal{C}'(\mathcal{S}))^2}$

- $T \gtrsim \frac{M^2}{\eta' \epsilon}$

- $r \gtrsim \frac{C^4 (T\eta')^2 M^2 \left(\mathcal{C}'(\mathcal{S})\right)^4 \log\left(\frac{C|\mathcal{S}|}{\epsilon \delta}\right)}{\epsilon^2} + d$

- *Zero initialized prediction layer*

- *Arbitrary $m$*

*Then, w.p. $\geq 1 - \delta$ over the choice of the initial weights, there is $t \in [T]$ such that $\mathbb{E}\,\mathcal{L}_{\mathcal{D}}(\mathbf{w}^t) \leq \min_{\mathbf{h} \in \mathcal{H}_{\mathcal{S}}^k,\, \|\mathbf{h}\|_{\mathcal{S}} \leq M} \mathcal{L}_{\mathcal{D}}(\mathbf{h}) + \epsilon$. Here, the expectation is over the training examples.*

We next consider ReLU activations. Here, $\mathcal{C}'(\mathcal{S}) = (\sqrt{32})^{\mathrm{depth}(\mathcal{S})} \sqrt{\mathrm{comp}(\mathcal{S})}$.

**Theorem 2.** *Suppose that all activations are the ReLU. Let $M, \epsilon > 0$. Suppose that we run algorithm 1 on the network $\mathcal{N}(\mathcal{S}, r, k)$ with the following parameters:*

- $\eta = \frac{\eta'}{r}$ *for* $\eta' \lesssim \frac{\epsilon}{(\mathcal{C}'(\mathcal{S}))^2}$

- $T \gtrsim \frac{M^2}{\eta' \epsilon}$

- $r \gtrsim \frac{(T\eta')^2 M^2 \left(\mathcal{C}'(\mathcal{S})\right)^4 \log\left(\frac{|\mathcal{S}|}{\epsilon \delta}\right)}{\epsilon^2} + d$

- *Zero initialized prediction layer*

- *Arbitrary $m$*

*Then, w.p. $\geq 1 - \delta$ over the choice of the initial weights, there is $t \in [T]$ such that $\mathbb{E}\,\mathcal{L}_{\mathcal{D}}(\mathbf{w}^t) \leq \min_{\mathbf{h} \in \mathcal{H}_{\mathcal{S}}^k,\, \|\mathbf{h}\|_{\mathcal{S}} \leq M} \mathcal{L}_{\mathcal{D}}(\mathbf{h}) + \epsilon$. Here, the expectation is over the training examples.*

Finally, we consider the case in which the last layer is also initialized randomly. Here, we provide guarantees in a more restricted setting of supervised learning. Concretely, we consider multiclass classification, when $\mathcal{D}$ is separable with margin, and $\ell$ is the logistic loss.

**Theorem 3.** *Suppose that all activations are $C$-bounded, that $\mathcal{D}$ is $M$-separable with w.r.t. $\kappa_{\mathcal{S}}$ and let $\epsilon > 0$. Suppose we run algorithm 1 on $\mathcal{N}(\mathcal{S}, r, k)$ with the following parameters:*

- $\eta = \frac{\eta'}{r}$ *for* $\eta' \lesssim \frac{\epsilon^2}{M^2 (\mathcal{C}(\mathcal{S}))^4}$

- $T \gtrsim \frac{\log(k) M^2}{\eta' \epsilon^2}$

- $r \gtrsim C^4 \left(\mathcal{C}(\mathcal{S})\right)^4 M^2 (T\eta')^2 \log\left(\frac{C|S|}{\epsilon}\right) + k + d$

- *Randomly initialized prediction layer*

- *Arbitrary $m$*

*Then, w.p. $\geq \frac{1}{4}$ over the choice of the initial weights and the training examples, there is $t \in [T]$ such that $\mathcal{L}_{\mathcal{D}}^{0-1}(\mathbf{w}^t) \leq \epsilon$*

### 3.1 Implications

To demonstrate our results, let us elaborate on a few implications for specific network architectures. To this end, let us fix the instance space $\mathcal{X}$ to be either $\{\pm 1\}^n$ or $\mathbb{S}^{n-1}$. Also, fix a bias parameter $1 \geq \beta > 0$, a batch size $m$, and a skeleton $\mathcal{S}$ that is a skeleton of a fully connected network of depth between 2 and $\log(n)$. Finally, we also fix the activation function to be either the ReLU or a $C$-bounded activation, assume that the prediction layer is initialized to 0, and fix the loss function to be some convex and Lipschitz loss function. Very similar results are valid for convolutional networks with constantly many convolutional layers. We however omit the details for brevity.

Our first implication shows that SGD is guaranteed to efficiently learn constant degree polynomials with polynomially bounded weights. To this end, let us denote by $\mathcal{P}_t$ the collection of degree $t$ polynomials. Furthermore, for any polynomial $p$ we denote by $\|p\|$ the $\ell^2$ norm of its coefficients.

**Corollary 4.** *Fix any positive integers $t_0, t_1$. Suppose that we run algorithm 1 on the network $\mathcal{N}(\mathcal{S}, r, 1)$ with the following parameters:*

- $\eta \lesssim \mathrm{poly}\left(\frac{\epsilon}{n}\right)$

- $T, r \gtrsim \mathrm{poly}\left(\frac{n}{\epsilon}, \log\left(1/\delta\right)\right)$

*Then, w.p. $\geq 1 - \delta$ over the choice of the initial weights, there is $t \in [T]$ such that $\mathbb{E}\,\mathcal{L}_\mathcal{D}(\mathbf{w}^t) \leq \min_{p \in \mathcal{P}_{t_0},\ \|p\| \leq n^{t_1}} \mathcal{L}_\mathcal{D}(p) + \epsilon$. Here, the expectation is over the training examples.*

We note that several hypothesis classes that were studied in PAC learning can be realized by polynomial threshold functions with polynomially bounded coefficients. This includes conjunctions, DNF and CNF formulas with constantly many terms, and DNF and CNF formulas with constantly many literals in each term. If we take the loss function to be the logistic loss or the hinge loss, Corollary 4 implies that SGD efficiently learns these hypothesis classes as well.

Our second implication shows that any continuous function is learnable (not necessarily in polynomial time) by SGD.

**Corollary 5.** *Fix a continuous function $h^* : \mathbb{S}^{n-1} \to \mathbb{R}$ and $\epsilon, \delta > 0$. Assume that $\mathcal{D}$ is realized[2] by $h^*$. Assume that we run algorithm 1 on the network $\mathcal{N}(\mathcal{S}, r, 1)$. If $\eta > 0$ is sufficiently small and $T$ and $r$ are sufficiently large, then, w.p. $\geq 1 - \delta$ over the choice of the initial weights, there is $t \in [T]$ such that $\mathbb{E}\,\mathcal{L}_\mathcal{D}(\mathbf{w}^t) \leq \epsilon$.*

### 3.2 Extensions

We next remark on two extensions of our main results. The extended results can be proved in a similar fashion to our results. To avoid cumbersome notation, we restrict the proofs to the main theorems as stated, and will elaborate on the extended results in an extended version of this manuscript. First, we assume that the replication parameter is the same for all nodes. In practice, replication parameters for different nodes are different. This can be captured by a vector $\{r_v\}_{v \in Int(\mathcal{S})}$. Our main results can be extended to this case if for all $v$, $r_v \leq \sum_{u \in \mathrm{in}(v)} r_u$ (a requirement that usually holds in practice). Second, we assume that there is no weight sharing that is standard in convolutional networks. Our results can be extended to convolutional networks with weight sharing.

We also note that we assume that in each step of algorithm 1, a fresh batch of examples is given. In practice this is often not the case. Rather, the algorithm is given a training set of examples, and at each step it samples from that set. In this case, our results provide guarantees on the training loss. If the training set is large enough, this also implies guarantees on the population loss via standard sample complexity results.

### Acknowledgments

The author thanks Roy Frostig, Yoram Singer and Kunal Talwar for valuable discussions and comments.

## Footnotes

[1] If $\ell$ is $L$-Lipschitz, we can replace $\ell$ by $\frac{1}{L}\ell$ and the learning rate $\eta$ by $L\eta$. The operation of algorithm 1 will be identical to its operation before the modification. Given this observation, it is very easy to derive results for general $L$ given our results. Hence, to save one paramater, we will assume that $L = 1$.

[2]That is, if $(\mathbf{x}, y) \sim \mathcal{D}$ then $y = h^*(\mathbf{x})$ with probability 1.

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
