[Reviews · NeurIPS 2017]

Reviewer 1



Proving the learnability of the stochastic gradient descent algorithm is an important task in deep learning. The authors consider this problem in a graphical framework in terms of computational skeletons and provide PAC learning type analysis. Since the problem itself is very difficult, the presented results are acceptable. The work is not at the top level to this reviewers due to the following two reasons: 1. The skeleton with the same number r of incoming neurons and the homogeneous linear kernel defined for the input node are pretty special. This leads to the proof of learnability of polynomials only, though Corollary 5 gives a special case of approximating a continuous function when it is realizable by the distribution. 2. The lower bound for r of order O(\epsilon^{-2} \log (1/\epsilon)) required for the accuracy \epsilon in Theorems 1 and 2 is very demanding.

Reviewer 2



This paper studies the problem of learning a class of functions related to neural networks using SGD. The class of functions can be defined using a kernel that is related to the neural network structure (which was defined in [Daniely et al. 2016]). The result shows that if SGD is applied to a neural network with similar structure, but with many duplicated nodes, then the result can be competitive to the best function in the class. Pros: - This is one of the first analysis for SGD on multi-layer, non-linear neural networks. - Result applies to various architectures and activation functions. Cons: - The definition of the kernel is not very intuitive, and it would be good to discuss what is the relationship between functions representable using a neural network and functions representable using this kernel? - The same result seems to be easier to prove if all previous layers have random weights and SGD is applied only to last layer. It's certainly good that SGD in this paper runs on all layers (which is similar to what's done in practice). However, for most practical architectures just training the last layer is not going to get a good performance (when the task is difficult enough). The paper does not help in explaining why training all layers is more powerful than training the last layer. Overall this is still an interesting result.

Reviewer 3



This paper addressed important theoretical problem how a standard stochastic gradient descend algorithm can guarentee on learning in polynomial time to approximate the best function in the conjugate kernel space of the network.It is claimed by authors that this is the first polynomial time guarentee for neural network learning with depth more than two. In general, the paper is clearly written but authors uses too many pages for background introduction such kernel classes and neural network learning,leaving little room for proofs of main theorems in the paper and thus make it quite challenging to fully understand all technical details. Another minor issue is the typo of descend. Authors keep using decend in the paper. Overall, I think this paper addresses an important theoretical problem on neural network learning.